# MAPF in 3D Warehouses: Dataset and Analysis

**Primary Keywords:** *(1) Applications; (7) Multi-Agent Planning;*

## Abstract

Recent works have made significant progress in multi-agent path finding (MAPF), with modern methods being able to scale to hundreds of agents, handle unexpected delays, work in groups, etc. The vast majority of these methods have focused on 2D "grid world" domains. However, modern warehouses often utilize multi-agent robotic systems that can move in 3D, enabling dense storage but resulting in a more complex multi-agent planning problem. Motivated by this, we introduce and experimentally analyze the application of MAPF to 3D warehouse management, and release the first open-source 3D MAPF dataset. We benchmark two state-of-the-art MAPF methods, EECBS and MAPF-LNS2, and show how different hyper-parameters affect these methods across various 3D MAPF problems. We also investigate how the warehouse structure itself affects MAPF performance. Based on our experimental analysis, we find that a fast low-level search is critical for 3D MAPF, EECBS's suboptimality significantly changes the effect of certain CBS techniques, and certain warehouse designs can noticeably influence MAPF scalability and speed.

## 1  Introduction

Multi-Agent Path Finding (MAPF) algorithms try to find a set of collision-free paths for multiple agents that minimize their aggregate cost, usually the sum of their travel times. These methods were initially motivated by computer games (Silver 2005). Since most computer games require agents traversing in 2D gridworlds, MAPF methods focused on these domains. In the last decade, the rise of large warehouses involving many robotic pick-up and drop-off agents has inspired more MAPF works that consider additional challenges in these tasks, as well as warehouse-specific 2D gridworld maps.

However, modern warehouses and robotic systems are no longer constrained to 2D navigation. Technologies like Attabotics[1], AutoStore Kardex[2], Retanus Robotics[3], all utilize robotic agents that travel in *3D* grids along rails. This enables dense and customizable storage at the expense of a more complex MAPF problem. Agents now have increased flexibility in planning by reasoning about the z-axis, which

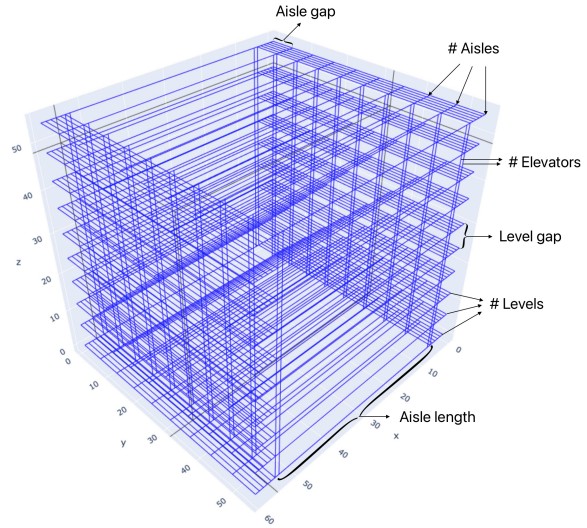

Figure 1: A 3D warehouse with six main map parameters: aisle length, number of aisles, aisle gap, number of levels, level gap, and number of elevators.

in turn leads to a substantially larger variety of paths and conflicts than are possible in 2D scenarios. Additionally, certain MAPF techniques that rely on 2D geometry are not applicable in a 3D environment. Given these changes, it is unclear how state-of-the-art MAPF problems will scale to such 3D warehouse environments.

Our objective is straightforward; analyze how MAPF methods work in realistic 3D warehouse scenarios. We try to address the following questions: How do MAPF methods scale in 3D? What are the unique challenges in 3D warehouses that are not explored in existing 2D literature? How does the geometry of the 3D warehouse affect performance?

Before we can start answering these questions, we first need to create realistic 3D scenarios. To this end, we obtained an approximate 3D warehouse vertex-edge schematic from an industry partner and created a parametrized 3D warehouse dataset that emulates real-world warehouses. Furthermore, unlike typical MAPF instances where agents just travel from start to goal locations, 3D warehouse agents typically travel from one bin location to a drop-off goal location and then back to their original position to return their bin. This returning step is an important distinction as

---

[1]https://www.attabotics.com/

[2]https://www.kardex.com/en/technology/products/autostore

[3]https://www.renatus-robotics.com/

it changes the difficulty of the task. These "start-goal-start" problems can be used to evaluate congestion at drop-off goal locations by having multiple agents have the same drop-off location, which occurs in practice. Succinctly, our main contributions are the following:

1. Creating a realistic open-source 3D warehouse domain that others can use.

2. Showing how certain warehouse geometries (e.g. aisle length) can significantly impact MAPF performance. Also evaluating the effect of goal congestion on MAPF performance.

3. Conducting a large ablation study of EECBS and discovering how suboptimality plays a big role in CBS's improvements' performance and how certain (EE)CBS improvements (e.g. Safe Interval Path Planning, Multi-Value Decision Diagrams) impact performance differently compared to existing 2D literature.

## 2 Related Work

### 2.1 Benchmark

The vast majority of MAPF methods are run on 2D grid-world environments. A non-comprehensive but representative list of MAPF works developed and tested in 2D environments are Conflict-Based Search methods (Sharon et al. 2015; Barer et al. 2014; Boyrasky et al. 2015; Li, Ruml, and Koenig 2021; Li et al. 2021), priority based methods (Erdmann and Lozano-Perez 1987; Ma et al. 2019; Li et al. 2022), and other MAPF solvers (Lam et al. 2022). In their corresponding experiments, these methods are all tested on MAPF scenarios which originate from the Moving AI benchmark by (Sturtevant 2012). These original Moving AI maps consist of several types of maps, notably city maps, computer game maps, and randomly generated maps that were then used for MAPF. Later on, warehouse grids were added to more closely replicate MAPF robotic planning in warehouse settings. Stern et al. (2019) describes these scenarios in detail, and notes that although there are different ways of assigning start-goal pairs, these scenarios contain randomly sampled start-goal pairs.

Recent research studies have also been extended to accommodate agents with large geometric shapes and volumes (Li et al. 2019b) or impose kinematic constraints over agents' move actions (Hoenig et al. 2017). These works consider variations in agents' size and actions where we consider a variation in the dimensionality of the environment.

To the best of our knowledge, our work introduces the first realistic (3D) warehouse dataset for MAPF. Our hope is that this dataset will spur new methods that handle complexities in 3D warehouses and serve as a benchmark similar to the existing 2D MAPF datasets.

### 2.2 MAPF and Start-Goal-Start Problems

The two most common MAPF problem variants are one-shot and life-long. In one-shot MAPF, we are given a graph and a set of agents with unique start and goal location. We must find paths for agents to reach their location that avoid obstacle collisions, vertex collisions (two agents at the same location), and edge collisions (two agents swapping locations at consecutive timesteps). When agents reach their goal location, they rest (without incurring cost) until the solution terminates when all agents reach their goal location. Lifelong MAPF has initial start-goal locations similar to one-shot MAPF. However, when an agent reaches its goal location, it is assigned a new goal location instead of resting. Since goals are constantly changing, lifelong MAPF planners almost always require receding horizon planning, i.e. they repeatedly re-plan conflict-free partial paths instead of full paths as in one-shot MAPF. This requires additional hyper-parameters on methods like partial-plan length (e.g. horizon) and re-plan frequency which affects performance. Lifelong evaluation also suffers from "starved" tasks where certain start-goal tasks are not completed, which is not allowable in real warehouse scenarios.

Start-goal-start problems are a simplified version of the multi-goal sequence MAPF problem introduced in Li et al. (2020) or multi-label sequence MAPF problem introduced in Grenouilleau, Hoeve, and Hooker (2021). We report results for standard start-goal problems as is typical. We additionally choose to include start-goal-start results as it is more realistic and evaluates the effect of goal congestion. We do not report life-long MAPF as it introduces several additional hyper-parameters that require tuning (our experimental section already includes plenty of hyper-parameters), and it has the task starvation issue which is not feasible in real warehouses. Future work could look into life-long MAPF performance in 3D warehouses.

### 2.3 Modern MAPF methods

There are several different state-of-the-art MAPF approaches that can be used to benchmark how existing methods perform on 3D scenarios, like EECBS (Li, Ruml, and Koenig 2021), BCP (Lam et al. 2022), and MAPF-LNS2 (Li et al. 2022). EECBS is a bounded sub-optimal method that employs the popular Conflict Based-Search framework with a focal low-level search and an explicit estimation high-level search. It additionally employs CBS improvements like Symmetry Reasoning (Li et al. 2021), Prioritized Conflicts (Boyarski et al. 2015), Bypassed Conflicts (Boyrasky et al. 2015), and others. Branch-and-Cut-and-Price (BCP) finds non-intersecting paths by using linear programming and the general branch-cut-price framework for reasoning over sets of paths. Both EECBS and BCP are complete and bounded-suboptimal methods. On the other hand, MAPF-LNS2 modifies prioritized planning (Erdmann and Lozano-Perez 1987) to find paths with minimum conflicts and then replans groups of conflicting agents to find a solution. Additionally, MAPF-LNS2 incorporated a version of Space-Interval Path Planning (SIPP) (Phillips and Likhachev 2011) which speeds up the low-level planner by reasoning away wait actions, and found this speeds up planning by $5\times$ compared to regular space-time planning. MAPF-LNS2 retains no theoretical guarantees but was found to outperform EECBS while having near-optimal solutions.

We chose to focus on EECBS and MAPF-LNS2 for our initial benchmark as this provides us with a modern bounded-suboptimal and non-bounded method. Running these methods on the 3D scenarios provides an accurate pic-

ture of how strong state-of-the-art MAPF methods scale to this new problem setting.

## 3  3D Warehouse Benchmark Suite

To ensure practicality and real-world accuracy, we collaborated with an industry-leading 3D warehouse company that provided us with an actual 3D warehouse schematic. Leveraging this resource, we derived six key attributes that parametrized the 3D warehouses: aisle length, number of aisles, aisle gap, number of levels, level gap, and number of elevators depicted in Figure 1.

**Aisle Length:** Aisles are depicted as elongated lines on the x-axis whose length is defined as the number of nodes.

**# of Aisles:** The total number of aisles in the x-y plane.

**Aisle Gap:** The y-axis spaces between adjacent aisles.

**# of Levels:** The number of stacked levels in the z-axis.

**Level Gap:** The spacing between levels along the z-axis.

**# of Elevators:** Elevators are depicted as vertical bars located at both ends of the aisles to interconnect levels along the z-axis. This sparse arrangement of elevators mimics the real-world map.

Besides the six main attributes, we also attach two dense grid structures at each end of the levels, which we refer to as margins. In real-world warehouses, they are used as buffer areas for robots to rest on while they are waiting for the previous agent to traverse the aisle, or used as placeholders for non-moving robots. We keep the dimension of each margin the same for every map.

We conducted experiments to identify the values for the six parameters that would yield similar results to the real-world map. Specifically, we found that the representative values are as follows: aisle length = 50, number of aisles = 10, aisle gap = 5, number of levels = 10, level gap = 5, and number of elevators = 2. We ran EECBS and MAPF-LNS2 on the real map and this representative parametrized map and validated that both algorithms produced similar results.

In order to investigate the diverse effects of different map configurations on MAPF algorithms, we expand our analysis by generating 14 additional maps. The first 12 maps are generated through systematic variations of individual parameters. For each of these maps, we selectively modify one of the six parameters, either doubling or halving its value, while keeping the remaining five parameters constant, following the specifications of the base map. This approach allows us to discern the distinct influence of each parameter on the MAPF algorithms' behavior. The last two maps are constructed by doubling or halving all six parameters.

By systematically generating these 14 additional maps, we ensure a comprehensive exploration of the varying effects that different 3D map configurations can have on MAPF algorithm performance. Additionally, this dataset provides valuable insights for understanding algorithm behavior under diverse and representative scenarios.

## 4  General Implementation Changes

We have two main sources of change that might require modifying EECBS or MAPF-LNS2. First, running MAPF on the 3D graph could require some algorithmic changes to the single-agent planners or conflict-resolution algorithms.

Instead of finding $(x, y, t)$ paths, we must find $(x, y, z, t)$ paths, which might seem like a combinatorially harder challenge. However, conceptually and implementationally, we can think of our 3D graph as just a generic graph with vertices $v$ with corresponding edges to neighboring vertices $v'$. In 2D, these vertices $v$ have an associated $(x, y)$, in 3D they now have an associated $(x, y, z)$. Regardless of 2D or 3D, we are still searching over a $(v, t)$ state-space and thus require no changes to the low-level or high-level search. Second, computing start-goal-start paths could require changing how the low-level planner works. This is easily done via small modifications to the state space and transition explained in the next section. We assume that edges take one unit timestep to travel regardless of their physical length.

### 4.1  Start-Goal-Start Paths

Most MAPF works just plan paths from start-goal locations. However, many real-world warehouse applications require the agent to pick up a bin from the start location, navigate to a drop-off location, and then return back to the start location to return the bin. We briefly describe how to change a MAPF planner for start-goal-start paths but note these changes are not novel and similar to changes in Li et al. (2020).

We want the MAPF planner to directly plan "start-goal-start" paths. This requires changing the state space from $(v, t)$ to $(v, t, i)$ where $i$ is a boolean indicating if we are traveling either from start to goal ($i = false$) or goal to start ($i = true$). The agent is not required to "hold" at the goal location, it instead holds at the end of its path (so it will rest at the start in the end).

This requires modifying the low-level planner for both EECBS and MAPF-LNS2 as it must plan paths that reach the goal and then navigate to the start. Implementationally, our agent starts at $(v_{start}, t = 0, i = 0)$ and must reach $(v_{start}, *, i = 1)$. $i$ can only be switched to 1 when the agent reaches $(v_{goal}, t, i = 0)$. Thus based on $i$, our search should go towards $v_{goal}$ or $v_{start}$. We can compute an accurate heuristic ignoring constraints by using a backward Dijkstra that starts at $(v_{goal}, *, i = 1)$.

A nice practical aspect of planning start-goal-start is that we can now easily control goal congestion by changing the number of agents that share the same intermediate goal. For example, we could have an "8 start-goal-start" problem where 8 agents share the same intermediate goal location. Controlling the number of agents sharing goals (e.g. to 4 or 16) allows us to manipulate (less or more) congestion in a way not as easily possible if we just planned start-goal paths. This also mimics real-world applications where robots usually have many start bins but few drop-off locations.

## 5  Conflict Based Search Changes

EECBS is a bounded sub-optimal MAPF solver that employs a high-level search over Conflict Tree (CT) nodes to resolve space-time conflicts in path proposals and a low-level search that proposes a new path for a single agent given CT node space-time constraints. By iteratively resolving conflicts in CT nodes, a goal CT node will provide a solution (if there exists) free of conflicts. Getting EECBS with

all its optimizations working in the 3D scenario with start-goal-start paths requires some modifications. Note that these modifications are mildly novel (as prior work with multi-goal MAPF did not use these optimizations), but are straightforward. MAPF-LNS2 does not require any additional modifications other than the low-level change described earlier.

## 5.1 Multi-Value Decision Diagram changes

Multi-Value Decision Diagrams (MDDs) are efficient data structures that store all paths of length $K$ between a start and goal, where $K$ is an input parameter based on the use-case (Sharon et al. 2013). For our start-goal-start purposes, the MDD is modified from finding start-goal paths of length $K$ to finding start-goal-start paths of length $K$ (searches in $(v, t, i)$ space). MDDs are powerful data structures that reason about sets of paths and are required for certain CBS improvements which we discuss in the next section.

## 5.2 CBS Improvements

Recent works have created several improvements that can speed up CBS. Several require modifications to work in the 3D setting as well as for start-goal-start path planning, while others can work out of the box.

**Directly usable.** Bypassing conflicts (Boyrasky et al. 2015): This technique reasons that instead of creating children $N'$ CT nodes of CT node $N$, we can instead replace $N$ with $N'$ if $N'$ resolves a conflict and has cost($N'$) ≤ cost($N$). Conceptually, this lets us produce smaller CTs and improve performance. This technique does not need to be modified and can be used out of the box.

Target reasoning (Li et al. 2021): Target reasoning occurs when an agent $I$ traverses over another agent $J$'s goal after agent $J$ reaches it and is resting at its goal location. Target reasoning applies a constraint on agent $J$ reaching its goal location before or after a specific time. Target reasoning is directly usable when applied to $(v_{start}, *, 1)$.

Corridor reasoning (Li et al. 2021): This method detects if agents are conflicting in a corridor (defined by a consecutive set of vertices with degree 2) and applies constraints on when the agents can enter the corridor. This method can directly be applied in the 3D workspace and start-goal-start paths without changes.

**Usable with changes to MDD.** Prioritized conflicts (PC) (Boyarski et al. 2015): PC determines which conflicts the CT node should decide to split on. It uses an MDD to determine if splitting on a conflict will increase the child CT node's cost and prefers such conflicts. Since we are able to generalize the creation of MDDs as described in section 5.1, we can use PC without further modifications.

Weighted Dependency Graph heuristic (WDG) (Li et al. 2019a): The WDG heuristic is an admissible heuristic for the high-level search. It computes pair-wise MDDs and uses an edge-weighted minimum vertex cover to compute the heuristic value. Again due to section 5.1, we can use this without further modifications.

**Not usable.** Rectangular reasoning (Li et al. 2021): Rectangular reasoning reasons for conflicts that occur in rectangular regions in 2D grid maps, and avoids a large number

| CBS Improvement | Is Effective in Common Intuition (2D) | Our Work (2D & 3D) |
|---|---|---|
| SIPP | Yes | Yes |
| Bypassing Conflict | Yes | Yes |
| Prioritized Conflicts | Yes | Depends on $w_{so}$ |
| Corridor Reasoning | Yes | Depends on $w_{so}$ |
| Target Reasoning | Yes | Yes |
| WDG | Yes | Depends on $w_{so}$ |

Table 1: A summary of which CBS improvements are beneficial. We found that EECBS's suboptimality ($w_{so}$) significantly affects the utility of certain improvements in both 2D and 3D. SIPP is much more useful in 3D than 2D scenarios.

of space-time conflicts by placing barrier constraints around the entire rectangular region. This method is unfortunately not usable in our 3D warehouse graph as the rectangular logic is not applicable as we can travel in the z-axis.

# 6 Experimental Results

We aim to answer the following questions:

1. How do modern MAPF methods (EECBS and MAPF-LNS2) scale to the 3D scenario as there are many more potential paths and conflicts?
2. How do EECBS and MAPF-LNS2 hyper-parameters affect performance?
3. How does goal congestion via start-goal-start problems affect performance?
4. How do map parameters affect performance?

## 6.1 Method Results and Analysis

As described in section 2, we specifically choose to use EECBS and MAPF-LNS2 as they are state-of-the-art bounded suboptimal and non-bounded methods respectively. Additionally, EECBS contains several Conflict-Based Search improvements that have been effective in 2D search but it is unclear how these will help in our 3D domain. We report median statistics across 10 random seeds, with each seed containing a unique set of start and goal locations sampled from a uniform distribution.

**EECBS** We examine the effectiveness of the six EECBS improvements: SIPP (Safe Interval Path Planning), Bypass/BP (Bypass Conflicts), PC (Prioritized Conflicts), CR (Corridor Reasoning), Target/T (Target Reasoning), and WDG (Weighted Dependency Graph high-level heuristic). A summary of their impact on performance under different suboptimalities is given in Table 1.

We choose to use two suboptimalities $w_{so} = 1.02$ (often used in the literature) and $w_{so} = 5$ in this study. We choose $w_{so} = 5$ for two reasons. First, existing unbounded methods, e.g. MAPF-LNS2 (Li et al. 2022) and LaCAM (Okumura 2022) compare against EECBS with $w_{so} = 5$ as a fair comparison (they deemed that suboptimality should be sufficiently high). Second, in real-life warehouse settings with many agents, Table 2 shows that a larger suboptimality is required for scalability and significantly faster planning time.

For a small EECBS suboptimality ($w_{so} = 1.02$), all six improvements demonstrate clear benefits as shown in Table

| EECBS Parameters | | Scenario Parameters | | | | | | | |
|---|---|---|---|---|---|---|---|---|---|
| | | Start-goal | | 1 start-goal-start | | 2 start-goal-start | | 8 start-goal-start | |
| $w_{so}$ | Improvements | Max # | Slowdown | Max # | Slowdown | Max # | Slowdown | Max # | Slowdown |
| 5 | Best (SIPP+T+BP) | 700 | 1 | 650 | 1 | 650 | 1 | 500 | 1 |
| | Best-SIPP | 500 | 22.77 | 250 | 37.19 | 300 | 20.46 | 200 | 44.49 |
| | Best-Bypass | 550 | 1.37 | 500 | 1.74 | 500 | 1.67 | 450 | 1.39 |
| | Best+PC | 550 | 1.27 | 550 | 1.20 | 600 | 1.19 | 450 | 1.07 |
| | Best+CR | 700 | 1.17 | 550 | 1.05 | 650 | 1.02 | 450 | 1.05 |
| | Best-Target | 500 | 1.21 | 500 | 1.48 | 500 | 1.38 | 500 | 1.33 |
| | Best+WDG | 600 | 1.26 | 500 | 1.28 | 650 | 1.25 | 450 | 1.17 |
| 1.02 | Best (All) | 170 | 1 | 70 | 1 | 50 | 1 | 30 | 1 |
| | All-SIPP | 130 | 1.30 | 60 | 1.14 | 40 | 1.52 | 30 | 1.94 |
| | All-Bypass | 130 | 1.00 | 70 | 1.07 | 40 | 1.06 | 30 | 0.86 |
| | All-PC | 130 | 0.84 | 50 | 0.95 | 50 | 0.51 | 30 | 0.88 |
| | All-CR | 110 | 1.02 | 70 | 0.64 | 40 | 1.07 | 30 | 0.20 |
| | All-Target | 100 | 2.44 | 50 | 1.12 | 40 | 1.04 | 30 | 0.35 |
| | All-WDG | 170 | 0.84 | 70 | 0.94 | 40 | 1.00 | 30 | 1.17 |

Table 2: We conduct an "add-or-substract-one" ablation study of CBS's improvements across different EECBS $w_{so}$ suboptimalities and scenario configurations. We choose the "Best" configurations (SIPP+Target+Bypass) and "All" improvements as the baseline for $w_{so} = 5$ and $w_{so} = 1.02$ respectively. We experiment with SIPP (Safe Interval Path Planning), Bypass/BP (Bypass Conflicts), PC (Prioritized Conflicts), CR (Corridor Reasoning), Target/T (Target Reasoning), and WDG (Weighted Dependency Graph high-level heuristic). For each configuration, we ran start-goal problems and start-goal-start problems with group size $1, 2, 8$. For each scenario, we ran a different number of agents in step sizes of 50 ($w_{so} = 5$) and 10 ($w_{so} = 1.02$), and stopped when they timed out (60 seconds) on over half their seeds (10 total seeds, needs to fail $\geq 5$ to stop). We report the largest number of agents with $\geq 50\%$ success rate in the "Max #" column, and the median relative slowdown compared to the corresponding baseline. Since each row contains adding/removing (+/-) certain parameters (e.g. Best-SIPP means SIPP was removed from Best, resulting in T+BP), a slowdown $> 1$ means that making this change hurts EECBS performance (as doing so slows EECBS). We highlight that the suboptimality significantly the impact of certain improvements (e.g. PC, WDG).

2 where the first row in entry $w_{so} = 1.02$ produces the best max # agent result, which is consistent with existing findings in 2D scenarios (Li, Ruml, and Koenig 2021). However, as the suboptimality is increased to 5, PC, CR, and WDG negatively impact runtime. We verified that this behavior also occurred in a 2D map (warehouse-10-20-10-2-1) when $w_{so}$ is changed from 1.02 to 5. Existing work has only explored the effect of CBS improvements in the low suboptimality regime, and our study of these parameters in the high suboptimality regime with the warehouse context leads to different optimal parameter settings. We encourage the reader to take a look at Appendix A which contains a comprehensive analysis of individual EECBS parameters and $w_{so}$. The only notable difference between 2D and 3D we found with $w_{so} = 5$ is that using SIPP in 3D yields a runtime speedup of $20\times$ and above while (Li et al. 2022) reports speedups of roughly $5\times$ in 2D scenarios. In general, this highlights that making careful choices regarding which parameters to activate becomes crucial in attaining optimal performance.

Our start-goal-start (s-g-s) results show that CBS improvements have a similar effect in start-goal-start across multiple group sizes and with regular start-goal problems. Additionally, we see that under $w_{so} = 1.02$, s-g-s problems are much harder than start-goal problems shown from a large difference in max # agents achieved between the start-goal and 1 start-goal-start column. We see goal congestion affects the scalability of EECBS shown by the decrease in max # agents across group sizes of 1, 2, and 8.

| Method | Start-goal | | 1 s-g-s | | 2 s-g-s | |
|---|---|---|---|---|---|---|
| | Max # | Slowdown | Max # | Slowdown | Max # | Slowdown |
| LNS2 | 1250 | 1 | 650 | 1 | 700 | 1 |
| LNS2- | 500 | 27.79 | 250 | 7.48 | 250 | 14.16 |
| EECBS | 700 | 1.80 | 650 | 1.22 | 650 | 1.19 |
| | 8 s-g-s | | 32 s-g-s | | 64 s-g-s | |
| LNS2 | 650 | 1 | 450 | 1 | 450 | 1 |
| LNS2- | 250 | 35.07 | 100 | 12.64 | 150 | 20.56 |
| EECBS | 500 | 1.19 | 400 | 1.04 | 450 | 0.86 |

Table 3: We compare LNS2 with SIPP (LNS2), LNS2 without SIPP (LNS2-), and EECBS $w_{so} = 5$ with the best improvements from Table 2 on different start-goal and start-goal-start with different group sizes $g$. The "Max #" column is the maximum number of agents where the method has a success rate $\geq 50\%$ across 10 seeds with a timeout of 1 minute. The "Slowdown" column is the relative runtime with the LNS2 row (e.g. a slow of 2.56 means it took $2.56\times$ longer to plan a path than LNS2). We see that removing SIPP dramatically slows down search and that EECBS performs around $1.8\times$ slower for easier problems but starts performing similarly to LNS2 when $g$ becomes large.

**MAPF-LNS2** Table 3 compares "LNS2" (MAPF-LNS2 with SIPP), "LNS2-" (MAPF-LNS2 without SIPP), and "EECBS" (EECBS with $w_{so} = 5$, SIPP, BP, and Target) across start-goal and $g$ start-goal-start problems. The "Max #" values are the maximum numbers of agents that the planner can solve in $\geq 50\%$ of the 10 seeds with a 1-minute timeout, while "Slowdown" is the median relative slowdown in runtime comparing against "LNS2". We first see that SIPP

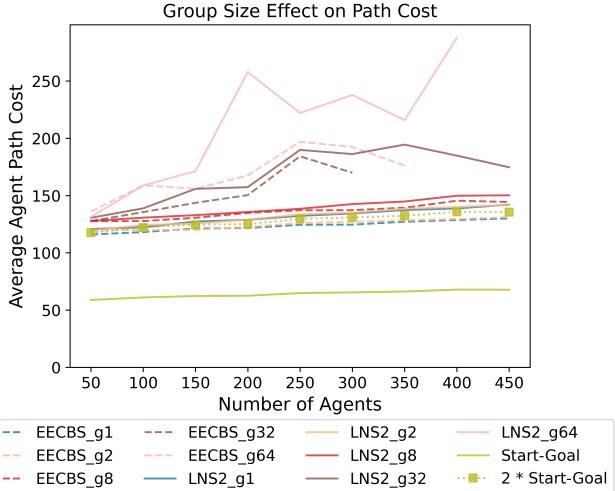

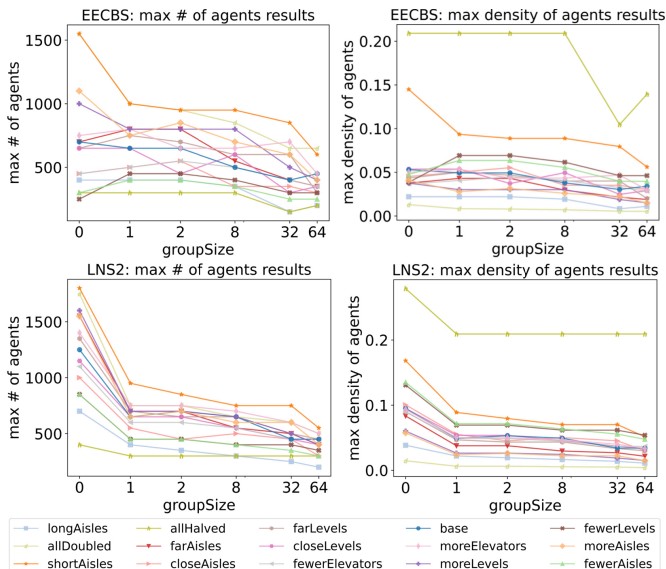

Figure 2: Our 3D scenarios have two different axes of change, the total number of agents as well as the start-goal-start group/congestion size $g$. We plot the average agent path cost of successful seeds when running EECBS (dashed) and LNS2 (solid) on different group sizes $g$. Each color corresponds to a unique group size. We plot the solution cost of start-goal problems in solid yellow, and $2*$ the cost in dotted yellow to provide context. We see that for low group sizes, e.g. 4, even 8, the path cost does not increase substantially and stays fairly close to the $2*$ expectation. However, as congestion at goals becomes large ($g \geq 16$), agents start taking large detours/waits causing the path cost to increase substantially. Comparing the dashed (EECBS) and solid (LNS2) lines within each color, we see LNS2's solution cost degrades as $g$ becomes large $g = 32, 64$, where LNS2 starts finding large suboptimal paths while EECBS is able to find significantly shorter paths.

Figure 3: We plot across different group sizes (0 denotes start-goal) among all maps. Each colored line represents a unique map, e.g. longAisles refers to the map where the aisle length is doubled. The left subplots show the absolute maximum number of agents achieved, and the right ones show the ratio of max # of agents to the total map size measured as the number of nodes.

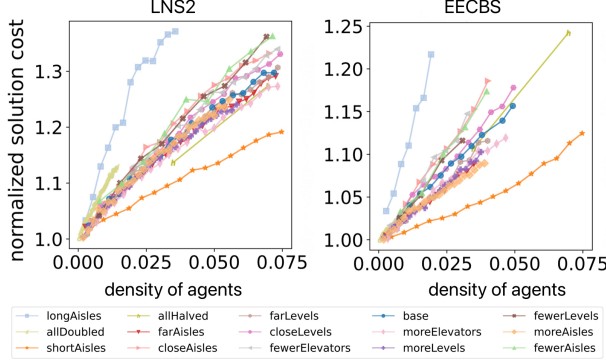

Figure 4: The y-axis of these figures represents the ratio of the final sum of cost to the initial sum of cost denoted as normalized solution cost. We only show the plots for group size 0 as it shows a similar trend with all other group sizes. The x-axis denotes the density of agents (# of agents over map size) and is capped at a density of 0.075. The different colored lines represent different maps. They clearly show that longer aisle length increases solution cost.

significantly speeds up performance in LNS2 by approximately $20\times$ on average. Additionally, we see that although EECBS is just $1.8\times$ slower on start-goal instances on solved start-goal problems, LNS2 is able to scale significantly better and solve 1250 agents while EECBS struggles after 700.

## 6.2 Goal Congestion Analysis

We investigate how goal congestion through $g$ start-goal-start problems (where $g$ agents go to the same intermediate goal location) differs from that of regular start-goal problems. Figure 2 depicts how the group size (the number of agents per intermediate goal location) impacts path length for EECBS and LNS2. We provide $2*$ the start-goal location as a comparison if start-goal-start paths simply copied start-goal paths on their way back. We see for small $g$ that this is roughly the case as they hug the $2*$ yellow dotted line. However, as $g$ increases $\geq 8$, both methods need to find longer solutions, suggesting that the problems are getting harder. Additionally, we see that LNS2 performs a little worse than EECBS for low $g$ and starts finding highly suboptimal paths for $g = 32, 64$. This highlights a possible drawback with using prioritized planning as LNS2's underlying low-level planner, which in these situations is forced to take long detours when congestion is high. This also highlights how start-goal-start problems can be practical tools in figuring out how different methods perform under different levels of congestion.

## 6.3 Benchmark Results

Figure 3 presents a comprehensive analysis of the scalability of the MAPF-LNS2 and EECBS algorithms across 15 maps by examining the maximum number of agents achieved from

| Change in Map Parameters | Map Size | EECBS Speedup | LNS2 Speedup |
|---|---|---|---|
| base | 1.0 | 1.0 | 1.0 |
| aisle length×2 | 1.38 | 0.48 | 0.31 |
| aisle length/2 | 0.81 | 2.00 | 4.63 |
| # aisles×2 | 2.05 | 0.97 | 1.85 |
| # aisles/2 | 0.48 | 0.72 | 0.84 |
| aisle gap×2 | 1.41 | 0.91 | 1.17 |
| aisle gap/2 | 0.75 | 0.79 | 1.03 |
| # levels×2 | 2.02 | 0.89 | 1.55 |
| # levels/2 | 0.49 | 0.78 | 0.61 |
| level gap×2 | 1.14 | 1.07 | 0.86 |
| level gap/2 | 0.92 | 1.03 | 1.19 |
| # elevators×2 | 1.14 | 1.21 | 1.82 |
| # elevators/2 | 0.93 | 0.96 | 0.72 |
| ∀ × 2 | 9.09 | 0.43 | 0.45 |
| ∀/2 | 0.11 | 0.22 | 0.29 |

Table 4: A summary of how different map parameters affect performance on 200-agent start-goal problems. The first column varies each of the six map parameters by doubling or halving it. The last two rows double and half all six parameters. The second column describes the relative change of the number of map nodes to that of the base map. The third and fourth columns show the runtime speedup for running EECBS and MAPF-LNS2 respectively.

different group sizes. To eliminate the confounding influence of varying map sizes, we present the data in two ways: the left figures employ the maximum number of agents as the y-axis, while the right ones use the maximum density of agents (i.e., the ratio of the maximum number of agents to the map size). We observe that the lines decrease sharply from group size 0 to group size 1, and a shallow slope after group size 1. This indicates that any start-goal-start problem is much harder than the goal-start problem, and congestion caused by larger group sizes does not hugely hinder performance compared to that caused by smaller group sizes.

Figure 4 shows the normalized solution cost (the ratio of the final sum of cost to the initial sum of cost) at different agent densities across all maps. Higher values mean that we need to find longer paths to avoid conflicts. It is interesting to note that the relationship between cost and agent density is roughly linear in most cases. Table 4 offers a perspective on the impact of map parameters on algorithmic runtime for 200 agents on start-goal problems. Note that the change in map size affects the agent density and can be a confounder of the relative speedup in relation to the base map.

Together these visualizations describe how the map configuration affects performance. Specifically, we see that aisle length has the biggest impact affecting speedup (Table 4), solution quality (Figure 4), and scalability (Figure 3). For any 3D warehouse designer, decreasing aisle length seems to boost performance across all metrics. Another interesting parameter is the number of elevators which noticeably impacts solution cost and speedup. This could imply that elevators are a computational bottleneck (e.g. many collisions

occur here) in the search algorithms.

Most existing work cast their focus on how different attributes of algorithms affect performance. We hope this analysis motivates future work to investigate manipulating warehouse structures to gain better performance.

## 7 Conclusion and Future Work

The main goal of our work is to create a realistic 3D warehouse benchmark and evaluate how existing MAPF methods perform across different realistic scenarios. We found that specific warehouse configurations exert a substantial influence on algorithmic performance. We additionally discovered that the suboptimality of EECBS has implications for the effectiveness of CBS-based enhancements. MAPF-LNS2 demonstrates strong scalability, yet faces challenges with solution costs under heightened congestion which we were able to manipulate via our start-goal-start scenarios that are frequently encountered in warehouse environments. We see many exciting directions to build off this work.

**Evaluating more MAPF methods and variants:** Recent methods like LaCAM (Okumura 2022) have shown impressive scalability on 2D environments at the expense of path cost. It is unclear how they will work in 3D warehouses with goal congestion. Likewise evaluating methods and hyperparameters for life-long MAPF could produce meaningful insights on their behaviour and shortcomings.

**Speeding up MDD creation:** Our results showed that constructing MDDs is a bottleneck for using certain CBS optimizations, and is not practical when scaling to large numbers of agents. Future work should figure out efficient ways to create MDDs or use them more selectively.

**Generalizing rectangular reasoning:** Rectangular reasoning could potentially be generalized to 3D geometry. Careful consideration is required as the 3D "rectangle" may not be fully connected as in our warehouses.

**Developing specialized techniques for handling goal congestion:** We saw that goal congestion can negatively affect performance, especially for MAPF-LNS2. Developing techniques like special constraints or developing conventions (e.g. always entering/leaving the goal along specific edges near the goal) could improve goal congestion performance.

**Warehouse structure optimization:** Our analysis showed how the aisle length and elevators have outsized impacts on performance. Additional warehouse structure like elevators in the middle could have non-trivial impacts. Directed edges could also decrease goal congestion degradation.

**Drop off location optimization:** Given a warehouse structure, optimizing the (goal) drop-off locations could reduce start-goal-start congestion and boost overall performance.

**Using Machine Learning for 3D MAPF:** Nearly all current machine learning approaches for MAPF have been designed for 2D environments. A crucial design in their models is inputting the graph as an image and using CNN architectures. A 3D warehouse renders CNNs useless and requires non-trivial consideration of what the inputs and network architecture should be. Additionally, it is possible that goal congestion is a specific task that learned models could exceed as they could potentially learn non-trivial congestion-avoiding behaviour and coordination.

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
