# OpenReview forum: "MAPF in 3D Warehouses: Dataset and Analysis"
_icaps-conference.org/ICAPS/2024/Conference — ICAPS 2024_

### Official Review · Reviewer_dJG8 · 2023-12-25

**Significance And Importance:** 3
**Soundness:** 4
**Novelty:** 3
**Clarity:** 4
**Overall Evaluation:** 2
**Confidence:** 3

**Weaknesses:**

2: No major or minor weaknesses.

**Contributions Of The Paper:**

The paper examines the performance of two families of MAPF methods in a 3D warehouse environment and reveals the effectiveness of each component in the existing MAPF methods under the new environment. These results also indicate directions for future work.

**Ethical Considerations:**

(1) Not Applicable: The paper does not have any ethical considerations to address

**Nomination For Best Paper:**

No

**Questions For Authors:**

1. how are the start and goal locations for each agent generated?
2. when varying group size, how is it decided which agent belongs to which group?

**Reproducibility:**

3: Authors describe the implementation and domains in sufficient detail.

**Strengths Of The Paper:**

The paper is well-written. The ablation study provides valuable insights for future research.

**Weaknesses Of The Paper:**

It is not clear how the 'start-goal-start' data are generated.
For example, how are the start and goal locations for each agent generated? When varying group size, how is it decided which agent belongs to which group?

---

> ### Author Rebuttal · Authors · 2024-01-27
>
> Posting a comment here for everyone to reduce repeated information across rebuttals.
>
> Thanks for your review! We have additionally created a public website https://mapf-16a81affbd82.herokuapp.com/ where anyone can view and download 3D warehouses (it takes a little time to load up initially). We will include additional features (e.g. a quick link to download the warehouses used in this study) and would appreciate input on additional features you think would help users.
>
> —————
>
> Q1: For regular start-goal problems, start and goal locations for agents are randomly sampled from vertices in the 3D warehouse with the constraint that no two agents should share the same start or goal location. It is possible for one agent's start location to be another agent's goal location however. Since our graphs are fully connected, there always exists valid start-goal paths. Note that existing 2D benchmarks also use start and goal locations randomly sampled from vertices as well.
>
> Q2: For start-goal-start problems, we arbitrarily assign agents to groups in a manner that groups G agents to the same intermediate goal location. Specifically, with a group size of G, we randomly number/order agents from 0 to N-1 and assign agents 0 to G-1 to goal location 1, agents G to 2*G-1 to goal location 2, etc. Mathematically we have ceil(N/G) goal locations with agent K getting assigned the K//G indexed goal location. These ceil(N/G) goal locations and N start locations are sampled similarly as in Q1, i.e. goals are randomly sampled from the vertices in the graph and unique across themselves, and starts are similarly picked.

---

### Official Review · Reviewer_kVkw · 2024-01-10

**Significance And Importance:** 2
**Soundness:** 2
**Novelty:** 2
**Clarity:** 2
**Overall Evaluation:** 1
**Confidence:** 4

**Weaknesses:**

0: Minor weaknesses requiring some work to be addressed for the paper to be accepted.

**Contributions Of The Paper:**

The paper introduces a benchmark set for 3D warehouses and uses this set to analyze modifications of two MAPF algorithms on this benchmark set.

**Ethical Considerations:**

(1) Not Applicable: The paper does not have any ethical considerations to address

**Nomination For Best Paper:**

No

**Questions For Authors:**

- Are elevators just paths along the Z-axis or are there actual elevator agents involved?
- You cited three different robotic 3D intra-logistic systems. Which one is closest to yours and do you think, you can abstract away from the different implementations?

**Reproducibility:**

3: Authors describe the implementation and domains in sufficient detail.

**Strengths Of The Paper:**

The interesting contribution of the paper is a generator for 3D benchmark scenarios for MAPF. An additional interesting contribution is an ablation study, where different MAPF algorithm optimizations are removed individually. Finally, the parameters for generating 3D scenarios are systematically varied in order to get an idea which parameter influences the algorithmic efficiency the most.

**Weaknesses Of The Paper:**

The main problem I had after reading the paper is that I had the feeling that I did not learn very much. Indeed, the concluding part in the final section is very short and general, and it does not make specific references to the fact that the scenarios are 3D instead of 2D. Indeed, after rereading the paper, it remains somewhat unclear what the influence of the fact that we deal with a 3D scenario has. There is actually one point, the application of rectangular reasoning, that is not applicable in 3D. But otherwise I did not see many differences to 2D MAPF. In fact, there is no evaluation on 2D scenarios. So it is impossible to see in how far the difference is significant.

Another weak point is the mechanics of the Z-axis. In Section 3, "elevators" are introduced, but it is not specified how agents use them. Is that just a path along the Z-axis or are there resource-limited elevators that carry agents? From the text and the demonstration in the supplement, one would expect the former interpretation. However, why would one call them elevators? This is never specified in the paper.

As a minor issue: i=true (line 259) and i=1 (line 266) may be the same in C, but not in C++ (or in language independent specifications).


I considered the replies in the rebuttal convincing and would suggest to address these questions in the final paper. For these reasons I upgraded my judgement to weak accept.

---

> ### Author Rebuttal · Authors · 2024-01-27
>
> Please read the top of dJG8's review.
>
> The main stated weakness is that there seems to be no substantial difference between 2D MAPF and 3D MAPF. We would like to argue that this is not a weakness but indeed one of the main findings from our analysis; MAPF methods previously only evaluated in 2D perform similarly in 3D warehouses. More specifically, the trends are similar in 2D and 3D (e.g. target reasoning helping EECBS, LNS2 scaling better than EECBS) but the absolute values can change (e.g. SIPP is 20x faster than space-time in 3D but only 5x faster in 2D). We actively looked for other differences in 2D vs 3D but did not find anything meaningful. MAPF in 3D domains have never been studied before. Thus although perhaps underwhelming (we agree more differences would be more exciting), this takeaway is novel and generally good news to the MAPF community as it shows that existing methods solely evaluated in 2D can generalize to new domains.
>
> We would also like to point out that we have several other novel findings!
> - Our findings about the effect of w_{so} suboptimality on EECBS's improvements is new and "unusual"; prior CBS-related works assume that combining all the improvements together is better.
> - Section 6.2's goal congestion analysis via start-goal-start problems is new and showcases how goal congestion affects performance independent of agent congestion in a manner no previous studies have tried before. The relative findings of goal congestion hurting performance are intuitive (and maybe "boring") but the study is novel and the nuances were not known beforehand (e.g. hurting LNS2 more than EECBS).
> - Section 6.3's analysis and results on the effect of 3D warehouse structure on MAPF is novel and not evident beforehand.
>
> Q1: Elevators are just edges/paths in the z-axis; maybe a more apt name is elevator-shafts or vertical-shafts. We will clarify it.
>
> Q2: The first link is the closest. Note our reference 3D warehouse (lines 53-56) was provided by a different company with a similar system. Different 3D warehouse implementations can be abstracted away successfully if they are axis-aligned (which is true in current real 3D warehouses). We believe our findings would be quite applicable to such scenarios across different implementations.
> Theoretically non-axis aligned set-ups can be modeled in existing MAPF frameworks using the generic (v,t) implementation (lines 239-243), although it is unclear how our 3D findings would translate to those domains.

---

### Official Review · Reviewer_Gy5C · 2024-01-22

**Significance And Importance:** 2
**Soundness:** 3
**Novelty:** 3
**Clarity:** 3
**Overall Evaluation:** 1
**Confidence:** 3

**Weaknesses:**

0: Minor weaknesses requiring some work to be addressed for the paper to be accepted.

**Contributions Of The Paper:**

This paper explores the application of Multi-Agent Path Finding (MAPF) in the context of 3D warehouse management, presenting the first open-source dataset for 3D MAPF. Leveraging this dataset, the study delves into distinctive characteristics of 3D MAPF, including considerations like aisle length in the warehouse and the impact of goal congestion. Existing 2D MAPF algorithms, specifically Enhanced Enhanced Conflict-Based Search (EECBS) and MAPF-LNS2, are adapted to the 3D environment, and their performance is systematically assessed.

**Ethical Considerations:**

(1) Not Applicable: The paper does not have any ethical considerations to address

**Nomination For Best Paper:**

No

**Questions For Authors:**

In your view, what sets apart a theoretical extension from 2D MAPF to 3D MAPF, and how does it differ from the extension presented in the current paper?

Would goal congestion essentially still be a significant factor?

**Reproducibility:**

4: Authors promise to release code and domains (whichever apply).

**Strengths Of The Paper:**

The creation of an open-source 3D warehouse dataset for MAPF, provides a valuable resource for researchers and enabling the exploration of algorithmic performance in complex 3D environments for the problem.

Another strength of the paper lies in its introduction of new features specifically tailored to 3D warehouses, such as aisle length and the impact of goal congestion. The exploration of start-goal-start settings adds relevance by simulating real-world scenarios and assessing their impact on MAPF algorithms. A study of these phenomena using the provided dataset enhances the understanding of algorithmic behavior in 3D warehouse environments.

The paper also offers a thorough evaluation of state-of-the-art 2D MAPF algorithms, namely EECBS and MAPF-LNS2, in the novel context of 3D warehouse environments. This comparison provides some valuable insights into the adaptability and performance of existing algorithms when applied to the challenges posed by three-dimensional spaces.

**Weaknesses Of The Paper:**

The paper's focus on extending MAPF to 3D warehouse management positions it as an application-oriented study rather than a purely theoretical model (meanwhile, I acknowledge the paper is an application one). While the authors explore the adaptation of MAPF algorithms from 2D to 3D environments, probably a more in-depth analysis of algorithmic foundations would strengthen the justification for this specific way extension of MAPF into the 3D domain.

I notice in particular the paper's discussion of the start-goal-start arrangement in 3D MAPF. The inclusion of start-goal-start introduces a departure from traditional 2D MAPF problems. This variation distinguishes the current problem, hence potential concerns about its relevance or justification should probably be addressed. The unique challenges presented by start-goal-start scenarios in 3D warehouses require a thorough exploration and explicit justification, specifically their impact on the practical application of existing 2D MAPF algorithms (which probably goes beyond what has been covered so far, in Sections 4/5 in the current paper).

---

> ### Author Rebuttal · Authors · 2024-01-27
>
> Please read the top of dJG8's review.
>
> Q1 and weakness 1:
>
> Short answer: Utilizing the different 2D vs 3D graph geometry and symmetries distinguishes theoretical MAPF extensions from 2D MAPF and 3D MAPF.
>
> Long answer: As described in Section 4 lines 230-241, a 3D search space and 2D search space can be represented identically by a (v,t) graph. Thus, purely theoretically, one could abstract away 2D vs 3D (i.e. they don't need to know which domain they are working on), and there is no distinguishing theoretical extension between the domains. However in practice, the differences between 3D warehouses and 2D maps are the distribution of vertices and connectivity across the two domains. Specific theoretical extensions for 2D specific MAPF techniques have leveraged 2D path symmetry. Thus 3D specific MAPF techniques need to leverage 3D symmetries, possibly for generalized rectangular reasoning (lines 513-516) or efficient MDD construction (lines 508-512).
>
> Q2: Our analysis of goal congestion showed that it significantly affects performance in 3D and 2D. We believe this is a problem that deserves specific attention regardless of the domain (lines 517-525).
>
> Weakness 2: Start-goal-start problems are required for real 3D warehouse operations where an agent picks up a bin, moves it to a goal location (where an object in the bin is usually removed/placed), and then needs to move the bin back to the original location (lines 57-61). We worked with an anonymous 3D warehouse company which specifically stated how they required agents returning bins to their original location and planning full start-goal-start paths instead of just start-goal paths. Thus, start-goal-start problems are motivated by real world 3D warehouse applications. The warehouse company also stated how goal congestion is a serious issue. Section 6.2 specifically analyzes goal congestion induced by start-goal-start problems. We also additionally explore goal-congestion in different methods/scenarios throughout the paper, as seen in Table 2, Table 3, and Figure 3.
>
> Lastly, viewing our start-goal-start experiments as a weakness implies that the paper would be stronger without start-goal-start results; we could remove them to the appendix if the reviewers' consensus is that it should be removed (please let us know), but we believe the additional insights about goal-congestion are meaningful. We can improve the justification in the camera-ready version as requested as well.

---

### Meta-Review · Area_Chair_GNgA · 2024-02-05

**Recommendation:** Accept (Oral)
**Confidence:** 4

**Metareview:**

The paper introduces a novel set of maps for benchmarking MAPF algorithms. The main novelty is that these maps represent 3D warehouses, where the agents can move across 3 spatial axes (x, y, z), which is in contrast with the commonly used MAPF benchmark (from Stern.et al, 2020), where the 2D grid maps are utilized. Additionally the authors suggest another way of creating a MAPF instance where each agent has to go from start to goal and then go back to start.

The authors present a comprehensive evaluation of the two state-of-the-art MAPF algorithms (MAPF-LNS2 and EECBS) on the introduced benchmark. An interesting finding is that certain enhancing techniques that speed-up algorithms in 2D do not have the same effect in 3D.

Overall, the paper provides a valuable resource for the MAPF community and an interesting discussion. Meanwhile, the algorithmic contribution is minimal and the number of methods evaluated (two) is not impressive. On the one hand, it would have been interesting to see the results of the optimal solvers (e.g. CBS), on the other, including into the evaluation solvers like PIBT/LaCAM may also be beneficial and increase the significance of the paper.

**Ethical Considerations:**

(1) Not Applicable: The paper does not have any ethical considerations to address